# VISOR: VIsual Spatial Object Reasoning for Language-driven Object Navigation

## Abstract

Language-driven object navigation requires agents to interpret natural language descriptions of target objects, which combine intrinsic and extrinsic attributes for instance recognition and commonsense navigation. Existing methods either (i) use end-to-end trained models with vision–language embeddings, which struggle to generalize beyond training data and lack action-level explainability, or (ii) rely on modular zero-shot pipelines with large language models (LLMs) and open-set object detectors, which suffer from error propagation, high computational cost, and difficulty integrating their reasoning back into the navigation policy. To this end, we propose VISOR (VIsual Spatial Object Reasoning) a compact 3B-parameters Vision–Language–Action (VLA) agent that performs human-like embodied reasoning for both object recognition and action selection, removing the need for stitched multi-model pipelines. Instead of raw embedding matching, our agent employs explicit image-grounded reasoning to directly answer "Is this the target object?" and "Why should I take this action?" The reasoning process unfolds in three stages: "think", "think summary", and "action", yielding improved explainability, stronger generalization, and more efficient navigation. Code and dataset available upon acceptance.

## 1 Introduction

Recent breakthroughs in Large Language Models (LLMs) and Vision Language Models (VLMs) have unlocked new possibilities in Embodied AI that were long considered infeasible. One such example is *language-driven* object navigation (Khanna et al., 2024; Yokoyama et al., 2024b), where an agent must locate a target object described in natural language. This task requires not only vision-language understanding, but also high-level reasoning for efficient navigation.

Existing approaches to language-driven object navigation generally fall into two main categories. The first is policy-based methods, typically end-to-end models trained with reinforcement learning (RL) or behavior cloning (BC) (Sun et al., 2025). These methods perform implicit reasoning by directly mapping embeddings to actions (Zhai et al., 2023). While efficient at inference, they are often task-specific, and struggle to generalize to novel environments.

Beyond learned end-to-end policies, pipeline-based methods assemble modular components in a zero-shot manner to handle perception, reasoning, and navigation (Gadre et al., 2023; Liu et al., 2025; Zhu et al., 2025). Proprietary VLMs (Hurst et al., 2024) are often integrated into these pipelines to conduct high-level reasoning, thereby offering greater explainability and stronger generalization in unseen scenarios (Ziliotto et al., 2025; Zhou et al., 2023). However, such methods suffer from error propagation across components and incur prohibitive inference costs, which hinder their deployment in real-world applications. Moreover, integrating LLMs explicit reasoning into navigation policies while balancing between efficiency and performance remains an open challenge (Taioli et al., 2025).

In this paper, we argue that the next generation of embodied agents should possess the CURE properties: *(i) Compact*, leveraging models with around 3B parameters, or fewer; *(ii) Unified*, implemented as a single Vision–Language–Action (VLA) model; *(iii) Reasoning-capable*, performing spatial reasoning from multiple observation sources (*e.g.,* the agent's POV RGB images, the instruction and a top-down map of the environment); *(iv) Explainable*, articulating both *what* they are doing and *why*.

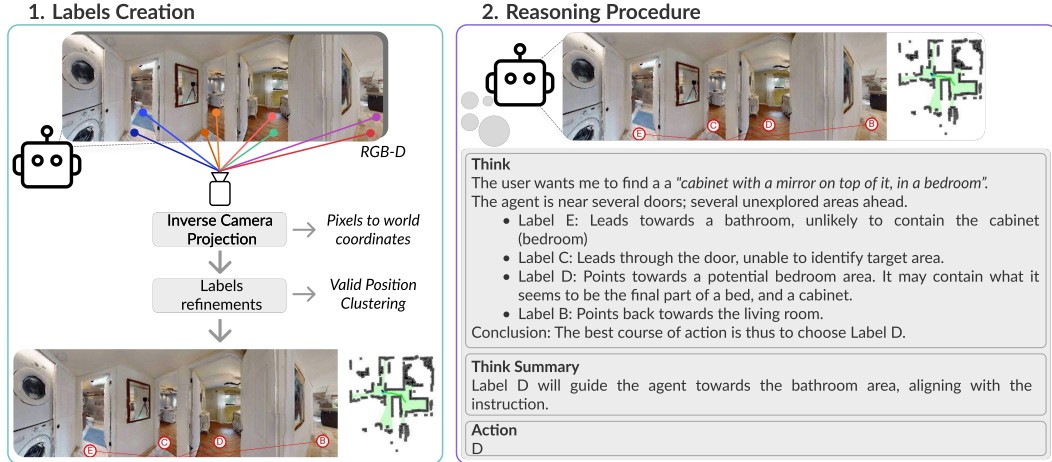

Figure 1: Given an instruction (*e.g., "cabinet with a mirror on top of it, in a bedroom"*), VISOR projects the panoramic observation into world coordinates via inverse camera projection. *Waypoint candidates* (Ⓔ, Ⓒ, Ⓓ, Ⓑ) are extracted through a clustering mechanism (Step 1) and superimposed on the panoramic view, serving as anchors for spatial reasoning. The *Reasoning traces* (Step 2) unfolds in three stages: `think`, `think_summary`, and `action`. Then, VISOR selects the most plausible label Ⓓ, projects it into world coordinates, and executes low-level actions via a shortest-path planner.

Based on these considerations, we introduce VISOR (VIsual Spatial Object Reasoning), a *compact*, *unified*, *reasoning-capable*, and *explainable* Vision–Language–Action (VLA) model for the language-driven object navigation task. Our 3B parameter model jointly reasons about object recognition and navigation, replacing raw embedding matching with explicit image-based reasoning that integrates the reasoning capabilities of LLMs with the perceptual grounding of VLMs.

At each inference step, the model produces three outputs, enclosed in distinct tags: *(i)* `<think>`, the detailed reasoning process; *(ii)* `<think_summary>`, a concise rationale that distills the key factors driving the decision; and *(iii)* `<action>`, the selected high-level waypoint. Importantly, `<action>` does not correspond to low-level navigation commands (*e.g., move forward, turn left*). Inspired by (Nasiriany et al., 2024; Goetting et al., 2024), candidate waypoints are projected onto the agent's observation and indexed with random labels (*e.g., F, E, 0*). The model selects the optimal waypoint, and a shortest-path planner then navigates to that location. To strengthen spatial reasoning, we extend the agent's field of view (FOV) with both panoramic RGB observations (768×256) and an online-built top-down map (256×256) of the environment (see Fig. 1). Mimicking the broad human horizontal field of view (HFOV) provides the agent with richer spatial context for decision-making.

Human-like reasoning is embedded directly into instance recognition, eliminating the need for multi-model pipelines (*e.g.,* object detectors). VISOR answers *"Is this the target object I'm looking for?"* while extending the same reasoning process to navigation, enabling the agent to justify each decision (*"Why am I taking this step?"*) within the `<think>` output. To enhance trustworthiness and explainability, a condensed rationale is provided within `<think_summary>` tags.

Training proceeds in two stages. First, we perform supervised fine-tuning (SFT) of Qwen 2.5 VL 3B (Bai et al., 2025) on our proposed *WAYS-Bench* dataset. Each sample includes a language instruction, panoramic and top-down RGB observations, the distance to the goal, a binary stop indicator, multiple candidate waypoints, the ground-truth waypoint (minimizing distance to the target), and reasoning traces from an LLM (Hurst et al., 2024). The dataset contains 36,170 training and 3,047 validation instances. Second, we perform RL post-training to further enhance its reasoning capabilities, increasing navigation efficiency. Code, datasets, and trained models will be released upon acceptance.

The contributions made with this paper can be summarized as follows:

- We present VISOR (VIsual Spatial Object Reasoning), a compact (3B) VLA model for language-driven navigation. Unlike prior multi-model pipelines, VISOR selects the most

suitable waypoint label directly from the agent's observation. The model is *unified* (a single model without external object detectors or segmentators), *reasoning-capable* (explicitly performing a thinking step before choosing an action), and *explainable* (articulating both what the agent will do and why). Code and dataset available upon acceptance.

- We introduce Waypoint Selection Bench (WAYS-Bench), the first dataset designed for supervised fine-tuning (SFT) in embodied waypoint selection. On top of that, we show that this dataset is well-suited for RL post-training, leading to improved navigation efficiency.

- We provide an extensive analysis of the limitations and failure cases of VISOR, offering key considerations for future improvements.

## 2 WAYPOINT SELECTION BENCH

To the best of our knowledge, no existing dataset provides waypoint-level supervision to support the training of reasoning-capable embodied navigation agents. To address this gap, we first outline the desired properties of such a dataset and then describe its construction. Specifically, each sample in our dataset is a tuple of: *(i) Target object*: the attributes of the target described in natural language; *(ii) Top-down map*: the position and trajectory of the agent, and the layout of the environment; *(iii) Panoramic observation*: the agent's observation from three cameras, each with a $90°$ horizontal field of view; *(iv) Waypoint candidates*: the available navigation positions in the observed image; *(v) Ground-truth label*: the waypoint that minimizes geodesic distance to the target; *(vi) Reasoning traces*: the rationale used to derive the action decision.

**Dataset desiderata.** To train our model effectively, the dataset should provide rich, context-aware description of the *Target object*, combining intrinsic attributes (*e.g.,* color, shape, material) with extrinsic attributes (*e.g.,* spatial relations, relative positions). These descriptions must encourage reasoning about target locations and include sufficient detail to disambiguate visually or semantically similar objects. In addition, the dataset should supply both the *Top-down map* of the environment and the *Panoramic observations* to facilitate spatial reasoning. Finally, each panoramic observation should contain *Waypoint candidates*, along with the *Ground-truth label* that minimizes geodesic distance to the *Target object*.

**Dataset Construction.** Our dataset is built upon GOAT-Bench (Khanna et al., 2024), which provides *Target object* references in various formats, including detailed natural language with intrinsic and extrinsic attributes. This serves as a strong source dataset, as the object descriptions, for example *"cabinet that is located near the bed and has a mirror on it"*, can support target location reasoning (*e.g.,* prioritizing navigation toward the bedroom) and include attributes that disambiguate similar objects.

We select episodes from GOAT-Bench that include natural language descriptions and apply an automatic filtering procedure to discard those with *non-unique* instructions, yielding a dataset where each episode is paired with a unique instruction. Next, we extend the field of view (FOV) of a shortest-path follower agent by equipping it with three RGB cameras and corresponding depth sensors.[1] Each onboard cameras (front, right, and left) provides a $90°$ horizontal field of view, together forming a *Panoramic observation*. An online *Top-down map* of the environment is constructed throughout the trajectory using implementation from Yokoyama et al. (2024a). Depth information is leveraged to identify valid navigation positions: each pixel is projected from *image space* to *world space* via inverse camera projection, and pixels exceeding the maximum depth threshold or corresponding to obstacles are discarded, thus yielding a segmentation mask of valid versus invalid positions.

We then apply DBSCAN clustering (Ester et al., 1996) to the valid positions and extract the corresponding centroids. Each centroid is projected back into *world space* as *Waypoint candidates*, and the one that minimizes the geodesic distance to the target position, computed using the Habitat simulator (Savva et al., 2019), is chosen as the *Ground-truth label*. Finally, the agent navigates toward the selected waypoint using Habitat's planner. This process repeats until the agent has a geodesic distance less than 1m to the target object.

For every navigation step, we save the following information: *(i)* the natural language instruction $\mathcal{I}$; *(ii)* the panoramic observation (RGB, $768 \times 256$); *(iii)* the top-down map (RGB, $256 \times 256$); *(iv)* the distance to the target goal; *(v)* the ground-truth label (*i.e.,* the waypoint with the minimum

---

[1]This setup can be readily simulated on humanoid robots or other embodied agents via head rotation.

geodesic distance to the target goal); *(vi)* the waypoint candidates (*i.e.,* distractors). Note that we can assign a random label to each waypoint, allowing the model to reason directly in image space. This reasoning style mimics the *"I should go here"* decision process. A sample of this process is shown in Fig. 1. Finally, we query GPT-4o (Hurst et al., 2024) to extract *Reasoning traces* for the supervised fine-tuning step. Specifically, we concatenate the panoramic and top-down images.We then provide the model with ground-truth information and generate reasoning traces using Chain-of-Thought (CoT), leading to the ground-truth label selection. Formally, we can define this dataset as $\mathcal{D}_{\text{SFT}}$. Dataset details are provided in Appendix Sec. B, and Tab. 1 summarizes the dataset statistics. Notably, each episode provides at most 1 stop action, while it can generate multiple non-stop actions.

Table 1: Dataset $\mathcal{D}_{\text{SFT}}$ statistics for train and validation splits. "Stop Actions" refers to samples where the correct action is `stop`; "Non-Stop Actions" refers to all other actions (*i.e.,* labels). "Avg. Action Space Size" indicates the average number of available actions per timestep.

| Split | #Samples | #Stop Actions | #Non-Stop Actions | Avg. Action Space Size |
|-------|----------|---------------|-------------------|------------------------|
| Train | 36,170   | 1,698         | 34,472            | 3.99                   |
| Val   | 3,047    | 131           | 2,916             | 4.10                   |

## 3 THE VISOR METHOD

Unlike prior work, VISOR is designed to be inherently explainable: at each step it reasons directly in the image space, provides a concise rationale for its decision, and selects the corresponding action. We begin by describe the agent setup, and then detail the architecture and training of VISOR.

**Agents Setup.** Language-driven object navigation requires an agent to locate a target object specified in natural language. We denote the language input by $\mathcal{I}$, representing either a high-level category or a detailed description of the target object. At the start of each episode, the agent is placed in an unknown environment under the continuous environment setting of the Habitat simulator (Savva et al., 2019). Given an instruction $\mathcal{I}$, the agent makes high-level decisions that specify waypoint in the environment. Labels are superimposed on the panoramic image to indicate the target waypoint the agent aims to reach, as described in Sec. 2. These high-level decisions are then translated into low-level actions using Habitat's shortest path planner. The action space is defined as $\mathcal{A} = \{$Forward 0.25m, Turn Right 15°, Turn Left 15°, Stop$\}$. Navigation ends when the agent issues the `Stop` action or after 500 steps. An episode is considered successful if the geodesic distance between the agent and the target object is less than 1m. This setup decouples high-level reasoning from low-level actions, enabling the model to focus learning on strategic decision-making while leveraging reliable navigation primitives.

**VISOR.** Within this setup, we introduce VISOR, which leverages a pre-trained VLM as the embodied navigation policy $\pi_\theta$. We design VISOR following the CURE properties. Thus, our objective is to unify perception, be reasoning-capable and explainable (*i.e.,* generate explicit reasoning traces within `<think>`), selecting the optimal waypoint label (`<action>`). We formulate language-driven object navigation as a label selection problem: at each inference step, the model selects the label that maximizes navigation efficiency. The model inputs follow Sec. 2, namely a panoramic RGB observation, a top-down environment representation, and the instruction $\mathcal{I}$. At inference time, we introduce a `Turn Around` action, which is absent from the training set. This action allows the agent to rotate by $180°$.

**Policy Warm-up.** Despite their impressive capabilities, state-of-the-art VLMs still exhibit systematic shortcomings (Tong et al., 2024), such as failures on simple queries and hallucinated responses. These issues are further amplified in compact VLMs (*e.g.,* $\sim$3B parameters), which often struggle to reliably follow long or precise instructions, adhere to system prompts, and generate well-structured outputs such as JSON or XML. To mitigate these limitations, we perform an initial supervised fine-tuning (SFT) stage. This step equips the policy $\pi_\theta$ with fundamental reasoning skills and aligns its outputs to our desired structured format. Formally, we perform SFT on $\mathcal{D}_{\text{SFT}} = \{\mathcal{P}^n, \mathcal{R}^n\}_{n=1}^N$, where $\mathcal{P}^n$ denotes the input prompt (comprising the system prompt, panoramic and top-down images and prompt, and instruction $\mathcal{I}$), and $\mathcal{R}^n = (r_1^n, \ldots, r_T^n)$ denotes the target output sequence of length $T$. The panoramic input includes both candidate labels (distractors) and the ground-truth label. To prevent overfitting to specific label names, we replace all labels with letters randomly sampled from the alphabet. Note that the labels are enclosed in a red circle, and superimposed onto the panoramic

image. The SFT objective is to maximize the likelihood of the target sequence under the policy $\pi_\theta$:

$$\mathcal{L}_{\text{SFT}}(\theta) = -\frac{1}{N} \sum_{n=1}^{N} \sum_{t=1}^{T} \log \pi_\theta(r_t^n \mid \mathcal{P}^n, r_{<t}^n). \tag{1}$$

This corresponds to standard teacher-forced cross-entropy training, where the model is optimized to autoregressively generate the ground-truth reasoning traces and structured outputs token by token.

**RL Optimization.** The effectiveness of modern LLMs and VLMs is largely attributed to their post-training stage. In particular, reinforcement learning (RL) optimization has recently been shown to substantially enhance reasoning abilities without requiring additional supervised data, *e.g.,* explicit reasoning traces $\mathcal{R}^n$. Among these methods, Group Relative Policy Optimization (GRPO, Shao et al. (2024); DeepSeek-AI et al. (2025)), and its sequence-based variant GSPO (Zheng et al., 2025), have emerged as a powerful approach for improving reasoning-based capabilities.

Following Zheng et al. (2025), we adopt the Group Sequence Policy Optimization (GSPO) objective:

$$\mathcal{J}_{\text{GSPO}}(\theta) = \mathbb{E}_{x \sim \mathcal{D}_{\text{RL}}, \{y_i\}_{i=1}^{G} \sim \pi_{\theta_{\text{old}}}(\cdot|x)} \left[ \frac{1}{G} \sum_{i=1}^{G} \min\left( s_i(\theta)\hat{A}_i, \ \text{clip}(s_i(\theta), 1-\epsilon, 1+\epsilon)\,\hat{A}_i \right) \right] - KL, \tag{2}$$

where the group-based advantage $\hat{A}_i$ is defined as:

$$\hat{A}_i = \frac{r(x, y_i) - \text{mean}\left(\{r(x, y_j)\}_{j=1}^{G}\right)}{\text{std}\left(\{r(x, y_j)\}_{j=1}^{G}\right)}, \qquad r(x, y) \in [0, 1], \tag{3}$$

and $r(x, y)$ is the reward function. The KL regularization term is defined as:

$$KL = \beta\,\mathbb{D}_{KL}[\pi_\theta \,\|\, \pi_{\text{ref}}], \tag{4}$$

where $\beta$ is the hyperparameter that controls the strength of the KL regularization, $\pi_\theta$ is the trainable policy and $\pi_{\text{ref}}$ is the reference policy.

Finally, the importance ratio $s_i(\theta)$ is computed from sequence likelihoods as:

$$s_i(\theta) = \left( \frac{\pi_\theta(y_i \mid x)}{\pi_{\theta_{\text{old}}}(y_i \mid x)} \right)^{\frac{1}{|y_i|}} = \exp\left( \frac{1}{|y_i|} \sum_{t=1}^{|y_i|} \log \frac{\pi_\theta(y_{i,t} \mid x, y_{i,<t})}{\pi_{\theta_{\text{old}}}(y_{i,t} \mid x, y_{i,<t})} \right). \tag{5}$$

Differently from GRPO, GSPO computes the importance ratio $s_i(\theta)$ at the sequence level rather than the token level, thereby aligning the optimization more closely with sequence-level rewards.

**Rewards.** To ensure that the generated output preserves the required structure with the three expected tags, we include a *format reward* that verifies the presence of `<action>`, `<think>`, and `<think_summary>` tags. Since $\pi_\theta$ has already been exposed to this format during the SFT stage, we assign a relatively small weight to this reward in order to avoid overemphasizing format compliance at the expense of other reward components.

Second, an *action reward* encourages the model to predict either the correct label or, when appropriate, the `Stop` action. Importantly, the `Stop` action is specified in the system prompt but not overlaid on the panoramic image. On the other hand, candidate labels are not included in the prompt and are only superimposed on the panoramic image. This design prevents the policy from relying solely on text-based shortcuts and instead forces it to visually "scan" the panoramic image to ground its decision in perception. By forcing image-based reasoning rather than from memorization of label, the agent develops stronger perceptual grounding. The complete prompt is provided in Appendix F.2.

**RL Dataset.** Directly applying RL post-training on $\mathcal{D}_{\text{SFT}}$ leads to reward hacking: the action distribution becomes biased toward label-selection actions while almost never producing the `Stop` action (see Tab. 1). Interestingly, the model still generates coherent rationales in the `<think>` and `<think_summary>` tags, but consistently fails to terminate episodes correctly.

To address this issue, we construct $\mathcal{D}_{\text{RL}}$, a balanced dataset derived from $\mathcal{D}_{\text{SFT}}$, where episodes are randomly sampled to enforce an equal ratio of `Stop` and non-`Stop` trajectories. Furthermore, unlike prior open-source implementations, we find that removing the Kullback–Leibler (KL) divergence regularization term (*i.e.,* setting $\beta$ to 0) significantly degrades performance and prevents policy convergence. We attribute this sensitivity to the relatively small model size considered in our setting.

## 4 EXPERIMENTS

### 4.1 EXPERIMENTAL SETUP

**Datasets.** We evaluate our method on two distinct benchmarks for language-driven object navigation: `InstanceObjectNav` (CoIN-Bench (Taioli et al., 2025)) and `ObjectNav` (OVON (Yokoyama et al., 2024b)). In the `InstanceObjectNav` task, the agent receives natural language descriptions of a *specific* target object (e.g., "the red mug with a white handle on the kitchen counter"), requiring fine-grained perception and spatial reasoning to identify it among visually similar objects. In contrast, the `ObjectNav` task only provides an object category as input (e.g., "Find the bed"), and the agent must locate any instances of that category. Therefore, we adopt CoIN-Bench, the more challenging dataset, as our primary experimental benchmark.

CoIN-Bench builds on episodes from GOAT-Bench (Khanna et al., 2024), applying automatic and manual filtering to ensure high-quality target visual observations and remove cases with 3D reconstruction errors or targets visually indistinguishable from distractors (*i.e.,* objects from the same category). Its split protocol follows GOAT-Bench, but with fewer validation episodes: 831 in `Val Seen`, 359 in `Val Seen Synonyms`, and 459 in `Val Unseen`. OVON is an open-vocabulary object navigation benchmark where the input is an open-set object category. Evaluation is conducted over three splits: `Val Seen` (categories observed during training), `Val Seen Synonym` (synonyms of seen categories), and `Val Unseen` (*novel categories*). Each split contains 3,000 episodes.

**Metrics.** For evaluation, following Anderson et al. (2018); Yadav et al. (2023), we report Success Rate (SR ↑) and Success weighted by Path Length (SPL ↑), defined as: $\text{SPL} = \frac{1}{N} \sum_{i=1}^{N} S_i \frac{l_i}{\max(p_i, l_i)}$, where $N$ is the number of episodes, $l_i$ is the shortest-path distance between the goal and the target in episode $i$, $p_i$ is the length of the trajectory, and $S_i$ is a binary indicator of success. While SR captures whether the agent stops at the correct target, SPL additionally accounts for path efficiency.

**Implementation Details.** We adopt Qwen 2.5 VL 3B (Bai et al., 2025) as the policy $\pi_\theta$. For SFT, we train the model on two NVIDIA H100 GPUs (80GB). We use a learning rate of $5 \times 10^{-5}$, a per-device batch size of $4$ with gradient accumulation of $4$, and train for one epoch. This stage requires $\sim 1.67$ wall-clock hours ($\sim 3.34$ GPU-hours). During RL training, the visual backbone is kept frozen for computational efficiency. Training is performed on $8\times$ NVIDIA A100-SXM (64GB) GPUs, while rollout (12 per step) is executed on a single NVIDIA A100-SXM (64GB) using the VLLM (Kwon et al., 2023). We use a learning rate of $1 \times 10^{-6}$ and a KL coefficient $\beta = 0.01$. RL training converges in $\sim 6.67$ wall-clock hours ($\sim 60$ GPU-hours).

**Baselines.** We are interested in evaluating methods following the CURE properties. To this end, we compare VISOR against SOTA `InstanceObjectNav` and `ObjectNav` *training-based* policies. For `InstanceObjectNav`, PSL (Sun et al., 2025) is trained on the ImageNav task ("go to this image") and then transferred to InstanceObjectNav, while Monolithic (Khanna et al., 2024) is an end-to-end RL policy designed for multimodal tasks. For `ObjectNav`, RL is a PPO-based policy (Wijmans et al., 2020) trained on OVON; BCRL is initialized with behavior cloning and fine-tuned with RL (Ramrakhya et al., 2023); and DAgRL is initialized with DAgger (Ross et al., 2011) and fine-tuned with RL.

Beyond these RL baselines, we include Uni-NaVid (Zhang et al., 2025a), a large-scale video-based VLA model trained on over 3.6 million navigation trajectories and designed to unify multiple embodied navigation tasks. It takes online RGB video frames and natural language instructions as input and directly outputs low-level actions. The base LLM is Vicuna-7B (Chiang et al., 2023), but its code is not publicly available. We also evaluate MTU3D (Zhu et al., 2025), which first extracts 2D features (using FastSAM (Zhao et al., 2023) for segmentation and a DINO backbone (Oquab et al., 2024) for feature selection) and 3D features (by projecting depth maps into point clouds), thereby enabling direct spatial memory. It then selects query objects or map frontiers and passes them to a trajectory planner, the Habitat shortest path planner.

### 4.2 EVALUATION RESULTS

**Models Relying Solely on Embeddings Fail to Generalize.** Methods that perform implicit reasoning, by directly mapping embeddings to actions, whether trained with RL or BC, struggle to generalize to unseen environments and objects. In Tab. 2, both the PSL (Sun et al., 2025) and Monolithic (Khanna

Table 2: Results on CoIN-Bench. We compare against Monolithic and PSL. We show the Oracle Stop (OS) performance, and the CURE properties (*C*ompact, *U*nified, *R*easoning and *E*xplainable).

| Method | PROPERTIES | | | | VAL SEEN | | VAL SEEN SYNONYMS | | VAL UNSEEN | |
|---|---|---|---|---|---|---|---|---|---|---|
| | C | U | R | E | SPL (↑) | SR (↑) | SPL (↑) | SR (↑) | SPL (↑) | SR (↑) |
| 1) Monolithic | ✔ | ✔ | ✗ | ✗ | 3.60 | 6.86 | 10.36 | 23.96 | 0.10 | 0.22 |
| 2) PSL | ✔ | ✔ | ✗ | ✗ | 6.74 | 15.88 | 7.99 | 24.23 | 2.67 | 8.50 |
| 3) VISOR (SFT) | ✔ | ✔ | ✔ | ✔ | 6.33 | 12.64 | 6.64 | 16.64 | 4.91 | 9.59 |
| 4) VISOR (GSPO) | ✔ | ✔ | ✔ | ✔ | **8.34** | 13.24 | **8.57** | 16.09 | **6.07** | 9.37 |
| 5) VISOR (GSPO)-OS | | | | | 10.93 | 16.37 | 19.58 | 27.51 | 8.54 | 11.49 |

Table 3: Results on OVON. We compare with RL, BCRL, DAgRL, Uni-NaVid, and MTU3D. We show the Oracle Stop (OS) performance and the CURE properties. † is using a 7B LLM, while ⋆ employs FastSAM and DINO features.

| Method | PROPERTIES | | | | VAL SEEN | | VAL SEEN SYNONYMS | | VAL UNSEEN | |
|---|---|---|---|---|---|---|---|---|---|---|
| | C | U | R | E | SPL (↑) | SR (↑) | SPL (↑) | SR (↑) | SPL (↑) | SR (↑) |
| 1) RL | ✔ | ✔ | ✗ | ✗ | 18.70 | 39.20 | 11.70 | 27.80 | 7.50 | 18.60 |
| 2) BCRL | ✔ | ✔ | ✗ | ✗ | 8.20 | 20.20 | 5.30 | 15.20 | 2.80 | 8.00 |
| 3) DAgRL | ✔ | ✔ | ✗ | ✗ | 21.20 | 41.30 | 14.40 | 29.40 | 7.90 | 18.30 |
| 4) Uni-NaVid † | ✗ | ✔ | ✗ | ✗ | 21.10 | 41.30 | **21.80** | 43.90 | **19.80** | 39.50 |
| 5) MTU3D ⋆ | ✔ | ✗ | ✗ | ✗ | **23.60** | 55.00 | 14.70 | 45.00 | 12.10 | 40.80 |
| 6) VISOR (SFT) | ✔ | ✔ | ✔ | ✔ | 9.69 | 22.83 | 10.50 | 24.78 | 8.61 | 21.80 |
| 7) VISOR (GSPO) | ✔ | ✔ | ✔ | ✔ | 12.48 | 21.70 | 11.49 | 19.82 | 11.86 | 22.00 |
| 8) VISOR (GSPO)-OS | | | | | 16.68 | 27.34 | 14.64 | 23.72 | 17.26 | 28.48 |

et al., 2024) baselines show a substantial drop in SR and SPL performance when moving from `Val Seen/Synonyms` to `Val Unseen` (rows 1, 2).This limitation is further evident in Tab. 3, where the best performing method, DAgRL, decreases from an SR of 41.30 to 18.30 (row 3) and MTU3D (row 5) also experiences a significant drop in SR on the `Val Unseen` split.

**Reasoning Helps Generalization.** From Tab. 3, we make two key observations: *(i)* Although VISOR reports lower performance than baselines on `Val Seen` and `Val Synonyms`, it does not exhibit large performance drops when evaluated on the `Val Unseen` split; *(ii)* The performance of VISOR remains consistent across splits, demonstrating robustness to both environmental changes and linguistic perturbations. While reasoning-based policies may underperform in familiar environments, their robustness in novel ones highlights a promising direction for scaling embodied agents. This trend is also observed in Tab. 2. Note that CoIN-Bench contains substantially fewer validation episodes (see Sec. 4). As a result, performance fluctuations should be interpreted with caution. In Figure 2, we show the reasoning capabilities of VISOR on a CoIN episode rollout.

**RL Post-Training Improves Navigation Efficiency.** We observe that RL training encourages the agent to predict the `Stop` action more frequently and with higher confidence, leading to more efficient trajectories. This improves overall navigation efficiency, as reflected in higher SPL scores, as shown in Tab. 2 and Tab. 3. However, we also find that the model occasionally terminates episodes prematurely, which can slightly reduce SR.

**Comparison with State-of-the-Art Policies.** Tab. 3 shows that our approach lags behind state-of-the-art policies in terms of raw performance. Nonetheless, it is important to highlight three distinguishing aspects of our method: *(i)* We train exclusively on detailed natural language descriptions rather than object categories, enabling richer and more flexible instruction grounding. *(ii)* Our method relies solely on the reasoning capabilities of a pre-trained VLM, without incorporating external modules

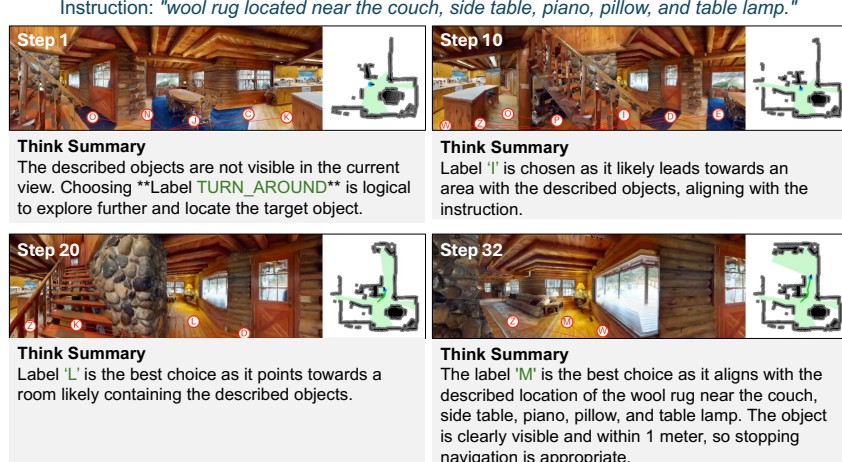

Figure 2: Reasoning capabilities of VISOR. At step 1, it selects a novel action. At Steps 10 and 20 it reason spatially to maximize navigation efficiency. Finally, at step 32 it successfully stop navigation, recognizing the same objects in instruction $\mathcal{I}$.

such as DINO-based visual embeddings (Oquab et al., 2024) and segmentator (Zhao et al., 2023) (row 5). *(iii)* Unlike row 4, we do not incorporate history but use a compact model with 3B parameters.

## 4.3 FURTHER ANALYSIS

A key feature of our model is its transparent reasoning process, available in both extended form (`<think>`) and summarized form (`<think_summary>`). We exploit this interpretability to examine representative failure cases, revealing limitations patterns in unsuccessful episodes.

**Hallucinations and Spatial Placement.** We observe that the model occasionally hallucinates labels (*i.e.,* reasons about labels that are not actually superimposed on the image) or misinterprets spatial placement, particularly left–right distinctions (*i.e.,* producing a plausible rationale for a label on the left when the label is in fact on the right). Robust spatial understanding remains an open research challenge (Kong et al., 2025). Samples are provided in Appendix C.1.

**Wrong Depth Understanding.** In this setup, agents are required to issue the action `Stop` when within 1m of the target object. However, accurate depth estimation in the absence of explicit depth observations remains an open research challenge. We further discuss in Appendix C.2.

**Markovian Setup.** Under the Markovian assumption, the agent's next action depends only on its current state. In the final stages of navigation (*i.e., last-mile navigation*), the agent may correctly identify and select a label that guides it toward the target. However, after moving in that direction, the agent may end up too close to the object to perceive it effectively, which can cause it to redirect incorrectly due to a lack of historical information. An illustrative example is provided in Appendix C.3.

**Variant Analysis.** During the early-stage design, we explored alternative strategies. While these variants were promising in principle, they proved difficult to train effectively and did not yield competitive performance (see Appendix D for detail).

While our agent achieves strong performance when navigating towards target objects, predicting the optimal label becomes challenging in the final steps of navigation (*i.e., last-mile navigation*). We further elaborate on the Oracle Stop experiment in Appendix E, and discuss potential future works in Sec. 6.

## 5 RELATED WORK

**Language-driven Object Nav.** Existing approaches tackle the tasks either with learned end-to-end policies or with zero-shot modular pipelines. Training-based methods rely primarily on RL (Sun et al., 2025; Khanna et al., 2024; Yokoyama et al., 2024b), leveraging vision–language embeddings for multi-modal representation (Radford et al., 2021; Zhai et al., 2023). In contrast, zero-shot

methods are built by composing multiple modules. For perception, open-vocabulary recognition is commonly achieved using open-set object detectors (Liu et al., 2025; Minderer et al., 2023), CLIP-based localization (Gadre et al., 2023) or a combination of 2D-3D features (Zhu et al., 2025). For navigation, classical frontier-based exploration (Yamauchi, 1997) is often employed, with recent work ranking frontier using VLMs (Yokoyama et al., 2024a; Li et al., 2023). A point-goal navigation policy then serves as a foundational component for reaching 3D world position (Wijmans et al., 2020). Finally, LLMs and VLMs are increasingly integrated into these pipelines to enhance perceptual grounding and reasoning capabilities (Ziliotto et al., 2025; Taioli et al., 2025; Zhou et al., 2023). In this paper, we leverage the best of both these paradigms by using a compact and unified model similar to the end-to-end trained methods but at the same time harnessing the reasoning capability of the large models used in the zero-shot methods.

**LLMs, VLMs and Embodied Reasoning.** Recent work has investigated integrating LLMs and VLMs into embodied agents to improve planning, reasoning, and interpretability. PIVOT (Nasiriany et al., 2024) frames embodied tasks as an iterative visual question answering. The idea is extended in Goetting et al. (2024) by incorporating depth information; however, both approaches rely on SOTA VLMs and require multiple inferences at each step, limiting efficiency. ReAct (Yao et al., 2023) interleaves language reasoning traces with task-specific actions, enabling agents to plan while acting. Huang et al. (2023) demonstrate that LLMs can maintain an inner monologue grounded in environmental feedback, enhancing planning performance in robotic control. SayCan (Ahn et al., 2022) grounds LLMs to real-world actions by combining language planning with value functions over pretrained low-level skills. Taioli et al. (2025) introduce an embodied Self-Questioner mechanism, where an onboard LLM and VLM iteratively generate and answer questions. Recently, hierarchical VLA models (Shi et al., 2025; Li et al., 2025; Xu et al., 2024) have been proposed to decompose high-level reasoning into low-level executable actions. MolmoAct (Lee et al., 2025) advances structured reasoning by producing three reasoning chains: Depth, for 3D reconstruction, Visual, for representing planned trajectories, and Action, for generating control commands. Finally, Gao et al. (2025) present an embodied generalist navigation agent designed around a "think-before-act" process. In this work, we enhance the reasoning capability of a VLM by incorporating reasoning traces from an LLM.

**Model's Post-training.** Step-wise reasoning, such as Chain-of-Thought (CoT) prompting (Wei et al., 2022), has been shown to substantially improve the ability of LLMs to perform complex reasoning. Toward this end, RL post-training has emerged as a key strategy to further enhance these capabilities. For example, the o1 model series (OpenAI et al., 2024) is trained with large-scale RL to elicit chain-of-thought reasoning. Similarly, the GRPO algorithm (Shao et al., 2024; DeepSeek-AI et al., 2025) demonstrates that outcome-based rewards can improve reasoning ability, even without preliminary SFT. Zheng et al. (2025) extend this approach by aligning token-level optimization with sequence-level objectives, thereby better matching the reward–outcome process. This training paradigm has recently been extended to VLMs and VLA models. For instance, Kimi-VL (Team et al., 2025) applies large-scale RL post-training to a VLM, while other works investigate RL post-training in embodied settings (NVIDIA et al., 2025; Wu et al., 2025; Zhang et al., 2025b), highlighting the growing relevance of reinforcement learning for reasoning and control in multimodal environments. We follow the post-training approach similar to GSPO (Zheng et al., 2025).

# 6 CONCLUSION

In this work, we propose VISOR, a 3B-parameter VLA agent that performs explicit image-grounded reasoning for both object recognition and action selection, which enables stronger generalization, better explainability, and more efficient language-driven object navigation. Several promising directions remain for future exploration. First, incorporating history of past observations and reasoning traces will extend our current setup to a non-Markovian framework, unlocking greater potential in SFT and RL. Second, depth information could be superimposed onto the top-down map, or extended to semantic map. Third, hallucinations remain a challenge for LLMs and VLMs. Incorporating rule-based rewards to penalize hallucinations presents an interesting avenue for mitigation.

# 7 REPRODUCIBILITY STATEMENT

To ensure full reproducibility of our research findings, we have taken several comprehensive measures. The complete source code, including all implementation details, along with the dataset used in this study, will be made publicly available in a GitHub repository upon acceptance. The detailed methodology for dataset generation is thoroughly documented in Section 2, including all parameters and procedures used. Furthermore, Section 4 provides comprehensive implementation details for policy training. We commit that VISOR will be fully open VLA with transparent training data, code, and recipe.

# 8 USE OF LLMS

We used an LLM during our dataset generation process for extracting reasoning traces (see Sec. 2 for detail). We also used an LLM to polish parts of the paper writing, such as rewording and improving grammar.

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

## A    APPENDIX SUMMARY

In this appendix, we provide the following:

- Section B: Dataset analysis.
- Section C: Discussion of failure cases.
- Section D: Variants of the proposed method and their analysis.
- Section E: We further discuss the Oracle Stop experiment.
- Section F: The prompts used throughout the paper.

## B    DATASET.

### B.1    $\mathcal{D}_{\text{SFT}}$

In this section, we present representative samples from the $\mathcal{D}_{\text{SFT}}$ dataset. Specifically, we show reasoning traces generated by GPT-4o (Fig. 3). The prompt used to collect these traces is provided in Appendix F.1. Moreover, in Fig. 4, we provide additional statistics on the distribution of natural language instructions in $\mathcal{D}_{\text{SFT}}$. Specifically, we report a boxplot of the instruction lengths, measured in tokens after tokenization with the Qwen2.5-VL tokenizer. The plot summarizes the variability across instructions and highlights the typical ranges encountered in the dataset.

Overall, instructions have a mean length of $21.16$ tokens and a standard deviation of $8.94$. The shortest instruction contains only 5 tokens, while the longest reaches 47 tokens. This indicates that most instructions fall within a compact range, but the dataset also includes a small proportion of longer instructions, which may require more complex grounding.

## C    FAILURE ANALYSIS

### C.1    HALLUCINATIONS AND SPATIAL PLACEMENT

The model often hallucinates when it comes to spatial reasoning. For example, in Fig. 5, we show an example where the agent hallucinate a label description, describing the corresponding target location as "kitchen" instead of a "living room". In Fig. 6, it ignores the spatial relationships mentioned in the instruction (*'mirror located to the right of the lamp and below the drawer'*) and selects the first object that matches the category as the goal ('mirror').

### C.2    WRONG DEPTH UNDERSTANDING

In Fig. 8, we show an example of the agent incorrectly estimates the distance to the target.

### C.3    MARKOVIAN SETUP

Another common failure case is when the agent incorrectly selects a waypoint leading away from the goal object, after moving too close to it (see Fig. 7 for example).

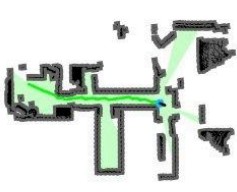

**GPT-4o Reasoning traces.**
<think>
The instruction describes a wardrobe located to the right of a bed, below a plant, and next to a chest of drawers. The wardrobe is a white dresser.
1. **Panoramic Image Analysis**:
- **Label C**: Leads to a room with a visible bed and furniture, but the wardrobe is not clearly visible.
- **Label X**: Shows a hallway with no visible furniture matching the description.
- **Label F**: Leads to a room where part of a bed is visible, suggesting proximity to the described location.

2. **2D Map Analysis**:
- The agent is positioned in a hallway with three possible directions. The map shows that label F leads further into a room, which is likely where the wardrobe is located.

3. **Instruction Alignment**:
 - The wardrobe is described as being to the right of the bed, suggesting it is in the same room as the bed. Label F is the most logical choice as it leads into a room where the bed is partially visible.

**STOP_NAV Consideration**: - The wardrobe is not clearly visible within 1 meter, so STOP_NAV is not appropriate. Based on the analysis, label F is the best choice to move towards the target object.
</think>
<think_summary>Label F is the best choice as it leads into a room where the bed is partially visible, aligning with the instruction that the wardrobe is to the right of the bed.
</think_summary>
<action>F</action>

Figure 3: GPT-4o reasoning traces. The input to the model is *"the wardrobe, which is located to the right of the bed. it is positioned below the plant and next to the chest of drawers. the wardrobe is described as a white dresser."* The distance to the goal is 2.81m.

## D  VARIANT AND ANALYSIS.

In this section, we provide two variants considered during the early-design of our approach.

### D.1  TWO-LEVEL SYSTEM: TOP-DOWN MAP PREDICTION

This variant is related to our main approach but differs in one crucial aspect. Instead of producing labels superimposed on the panoramic image, it sought to exploit the top-down map more explicitly to enhance spatial reasoning. We adopted a similar setup, except that the input consisted of three separate RGB images rather than a single panoramic image, while the top-down map remained unchanged. The agent was trained to directly predict the next goal location by specifying a displacement on the map relative to its current position (*e.g.,* "two cells to the left and one cell down"). This displacement was then projected into world coordinates, and the low-level navigation policy executed the trajectory toward that point.

In practice, however, reliably regressing the robot's displacement on the top-down map proved difficult, and training convergence remained unstable.

### D.2  REGRESSION OVER THE PANORAMIC OBSERVATION

We follow a similar approach as in Appendix D.1, but with two key modifications: *(i)* the three separate input images are replaced with a single panoramic image; and *(ii)* the prediction is made directly on the panoramic image rather than on the top-down map, effectively allowing the agent to point to its intended target location (*i.e.,* mimicking *"I want to go here <x, y>"*).

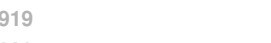
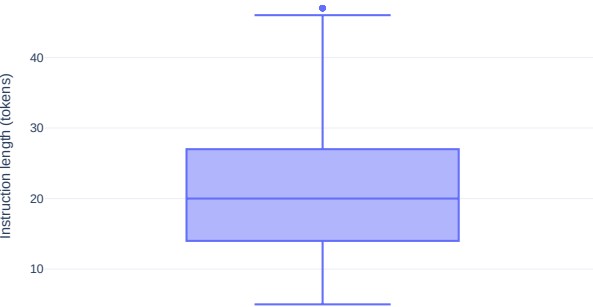

Figure 4: Boxplot of instruction lengths (in tokens) for the $\mathcal{D}_{\text{SFT}}$ dataset. Overall, instructions have a mean length of 21.16 tokens, requiring complex grounding.

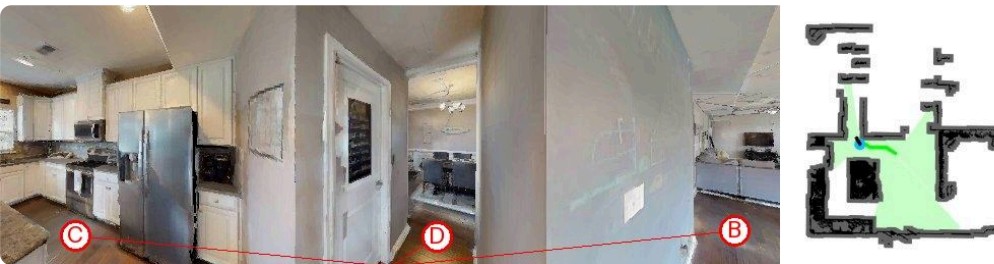

**<think>** The instruction specifies looking for a *"cabinet with a chalkboard on it,"* located *"to the right of the curtain."*
  1. Panoramic Image Analysis:
    • The image shows a kitchen area with cabinets.
    • There is a door leading to another room, which might contain the target cabinet.
    • The cabinet with a chalkboard is not immediately visible in the current view.
  2. 2D Map Analysis:
    • The agent is positioned near the kitchen, facing towards the door.
    • The map indicates unexplored areas beyond the current field of view, suggesting potential locations for the cabinet.
  3. Label Consideration:
    • Label D: Leads directly through the door, potentially into the next room where the cabinet might be located.
    • Label B: Points towards the kitchen area, unlikely to contain the cabinet described.
    • Label C: Points back towards the kitchen, similar to B.
    • STOP_NAV: The cabinet is not visible within 1 meter, so stopping is not appropriate.
  4. Conclusion:
    • The best course of action is to choose Label D, as it leads towards the unexplored area where the cabinet might **be located.**
**</think>**
**<think_summary>** The cabinet with a chalkboard is likely in the next room. Choosing Label D will guide the agent towards this area, aligning with the  instruction.
**</think_summary>**
**<action>** D **</action>**

Figure 5: **Failure due to hallucination**. The input description is *"cabinet with a chalkboard on it. the cabinet is located to the right of the curtain."* The hallucination is highlighted in red. In particular, label B does not guide the agent toward the kitchen; instead, it leads the agent to explore the living room area.

We trained this variant with SFT followed by GRPO, using a custom reward design. In particular, we experimented with two reward types: *(i)* discrete target rewards (*i.e.,* 0 and 1), where the agent was rewarded if its predicted location $(x, y)$ fell within a small threshold of the ground-truth coordinate;

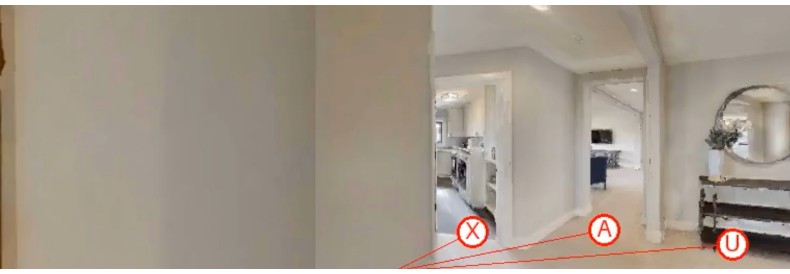

**<think>** The instruction specifies that the target object is a mirror located to the right of a lamp and below a drawer. In the panoramic image, there is a mirror visible on the right side. The labels available are 'X', 'A', and 'U'.
-Label 'X' is positioned towards the mirror, which matches the description provided in the instruction.
- Labels 'A' and 'U' do not seem to be directly associated with the mirror.
Given the proximity of the mirror to the agent's current position and the clear visibility of the mirror, it is reasonable to conclude that the object is within 1 meter and clearly visible. Therefore, the most efficient action is to stop navigation.
**</think>**

**<think_summary>** The mirror is clearly visible and within 1 meter, matching the instruction. The best action is to stop navigation.
**</think_summary>**

**<action>** STOP_NAV **</action>**

Figure 6: **Failure due to Incorrect spatial understanding.** The input description is "*mirror located to the right of the lamp and below the drawer*". However, the agent incorrectly stops near a mirror, by completely ignoring the spatial relationship with other objects in the instruction.

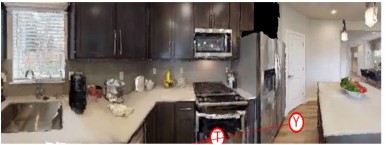 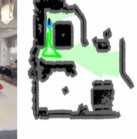 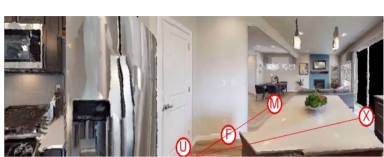 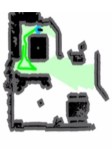

**Think Summary**
The refrigerator is visible but not within 1 meter. Choosing label 'Y' will guide the agent towards the refrigerator efficiently.

**Think Summary**
Label 'F' is the best choice as it directs the agent towards the kitchen area where the refrigerator is likely located.

Figure 7: **Failure due to Markovian setup.** The agent correctly reasons about the location of the 'refrigerator' (left). However, after reaching the predicted waypoint, the agent forgets the fact that it had already seen the 'refrigerator' in previous steps, and makes an incorrect reasoning (right). Such failures happen due to the underlying Markovian assumption in our method.

and *(ii)* continuous rewards, where the reward scaled inversely with the distance between the predicted and ground-truth coordinates. Again, training proved difficult, and the model was not able to converge.

# E  ORACLE STOP

In the main paper, we provide an "Oracle Stop" experiment. To do that, at each simulator step, we calculate the distance from the agent position to the target object position. If less than 1m, we override the `Stop` action.

From the results reported in Table 2 and Table 3, we can estimate performance of the *current* model, while having perfect decision making performance of the `Stop` action. As we can see, especially in

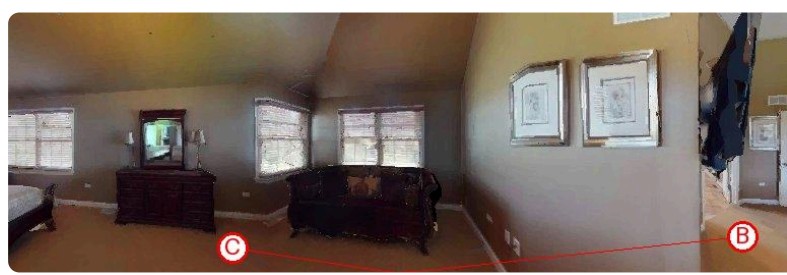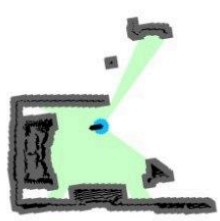

> **<think>**
> The instruction specifies a "cabinet that is located near the bed and has a mirror on it." In the panoramic image, there is a cabinet with a mirror visible on the left side. This matches the description provided in the instruction.
> The top-down map shows the agent's current position and orientation. The agent is facing towards the area where the cabinet is located. The labels available are 'C', 'B', and 'TURN_AROUND'.
> Label 'C' is directly aligned with the cabinet and mirror, indicating that the target object is within close proximity. Given the proximity and visibility of the cabinet, it is reasonable to conclude that the object is clearly visible and within 1 meter.
> Since the object matches the description and is clearly visible, the most efficient action is to stop navigation.
> **</think>**
>
> **<think_summary>**
> The cabinet with a mirror is clearly visible and within 1 meter, matching the instruction. Therefore, the appropriate action is to stop navigation.
> **</think_summary>**
>
> **<action>** STOP_NAV **</action>**

Figure 8: **Failure due to Wrong depth understanding**. The input description is *"a cabinet located near the bed with a mirror on it."* However, the agent incorrectly estimates the distance to the target as being less than 1m.

Table 3, the Oracle Stop diminish the discrepancy between the GSPO model and the best performing model, Uni-NaVid (SPL of 17.26 vs 19.80).

## F  PROMPTS

### F.1  SFT DATASET CONSTRUCTION (GPT-4O)

The following snippet shows the system prompt used to construct the supervised fine-tuning (SFT) dataset $\mathcal{D}_{\text{SFT}}$ with GPT-4o.

```
def build_prompt(sample, label_list, gt_label):
    action = "STOP_NAV" if sample["should_call_stop"] else gt_label
    return (
        "You are reasoning spatially and visually for an embodied agent tasked with locating an object
        ↪  described in natural language.\n\n"
        "The agent receives:\n"
        "1. A **768x256 panoramic RGB image** (current field of view)\n"
        "2. A **256x256 top-down 2D map** (explored area, obstacles, agent's position and heading)\n"
        "3. A natural language **instruction** describing the target object\n"
        "4. A list of **labels** corresponding to the coordinates in the panoramic image\n"
        "##
        "Your task: Determine the label to choose that will most efficiently bring the agent to the target
        ↪  object. Use visual cues, spatial reasoning, and common sense..\n\n"
        "Think step by step. Your reasoning must be clear, grounded, and structured.\n\n"
        "**Prediction rules:**\n"
        "- If the object is **clearly visible and within 1 meter**, set '<action>STOP_NAV</action>\n"
        "- Otherwise, '<action>best label</action> and select the best labels using the rules explained
        ↪  above. \n\n"
        "##
        "##
        "You are given internal data to guide your output:\n"
        f"<GROUND TRUTH DATA> instruction: {sample['instruction']}, should_call_stop:
        ↪  {sample['should_call_stop']}, available_labels: {label_list}, label_to_choose:{gt_label}
        ↪  </GROUND TRUTH DATA>\n\n"
        "Use this data to inform your reasoning, but **do NOT mention or refer to it explicitly**.\n"
        "Your reasoning and predictions must appear as your own, based on the scene and instruction.\n\n"
        "<think>\n"
```

```
       "...Detailed internal reasoning based on the panoramic image, map, and instruction.\n"
       "Consider each label logically. Justify if and why STOP_NAV is appropriate.\n"
       "</think>\n"
       "<think_summary>\n"
       "...Concise summary of decision and reasoning...\n"
       "</think_summary>\n"
       f"<action>\n{action}\n</action>"
    )
```

## F.2 RL POST-TRAINING

The following snippet shows the prompt used during RL post-training.

```
conversation = [
    { ### System Prompt
        "role": "system",
        "content": [
            {
                "type": "text",
                "text": "You are a helpful and intelligent embodied navigation agent that provides
                ↪  well-reasoned and detailed responses. First, reflect internally through a detailed
                ↪  reasoning process. Then, summarize your reasoning, and finally predict the action.
                ↪  Always respond using the exact format below: <think>\n...your internal
                ↪  reasoning...\n</think>\n<think_summary>\n...a brief summary of your
                ↪  reasoning...\n</think_summary>\n<action>\n...exactly ONE of the available labels or
                ↪  STOP_NAV.\n</action>",
            }
        ],
    },
    {
        "role": "user",
        "content": [
            ### Panoramic Observation
            {
                "type": "image",
                "image": sample["observation"],
                "resized_height": 256,
                "resized_width": 768,
            },
            {
                "type": "text",
                "text": " This is a panoramic observation of the agent's POV. It shows the current
                ↪  environment from the agent's point of view.",
            },
            ### Top-down Map
            {
                "type": "image",
                "image": sample["top_down_map"],
                "resized_height": 256,
                "resized_width": 256,
            },
            {
                "type": "text",
                "text": " Top-down map (green = explored space, gray = border and obstacles, blue circle =
                ↪  robot position (black line = robot heading). No objects are shown here, only this
                ↪  information.",
            },
            {
                "type": "text",
                "text": (
                    f"The instruction given to the agent is: '{sample['instruction']}'.** \n "
                    f"**Your task**: Determine the label that most efficiently guides the agent to the
                    ↪  target object. Use visual cues, spatial reasoning, and common sense.\n "
                    #
                    "Reason visually (panoramic) and spatially (top down map) to decide:\n "
                    "- If the object matches the description, it is clearly visible and it's under 1 meter,
                    ↪  the action must be STOP_NAV. The top-down map is the most reliable for estimating
                    ↪  distance and object proximity. The agent is shown as a blue dot, and
                    ↪  obstacles/objects are in black. \n "
                    #
                    "- Otherwise, use reasoning and common sense to select **ONE** of the available labels
                    ↪  in the panoramic image where the agent should move toward to reach the target
                    ↪  object. They are red circled. \n\n "
                    #
                    f"Think step by step. Your reasoning must be structured, clear, grounded in all labels
                    ↪  available on the image (red circles) and the instruction."
                    f"Select an action after the thinking process.  \n "
                ),
            },
```

```
        ],
    },
]
```

