# OpenReview forum: "VISOR: VIsual Spatial Object Reasoning for Language-driven Object Navigation"
_ICLR.cc/2026/Conference — Submitted to ICLR 2026_

### Official Review · Reviewer_oKXw · 2025-10-27

**Soundness:** 2
**Presentation:** 3
**Contribution:** 3
**Rating:** 4
**Confidence:** 4

**Summary:**

This paper introduces VISOR (VIsual Spatial Object Reasoning), a novel framework for language-driven object navigation embodied in a compact, unified 3B-parameter Vision-Language-Action (VLA) model. The work identifies key limitations in two dominant paradigms: (1) end-to-end policy-based methods that struggle with generalization and explainability, and (2) modular, pipeline-based methods that suffer from error propagation and high inference costs. VISOR proposes an elegant middle ground by reformulating the navigation task as an explicit, image-grounded reasoning problem. Instead of predicting low-level actions, the agent selects from a set of high-level waypoint candidates projected directly onto its panoramic visual observation. The core of the method is a three-stage generative process—"think", "think summary", and "action"—which forces the model to articulate its reasoning before making a decision. To train this model, the authors introduce WAYS-Bench, a new dataset built upon GOAT-Bench, which provides supervision for waypoint selection and includes reasoning traces generated by a large-scale LLM. The model is trained in two stages: Supervised Fine-Tuning (SFT) followed by Reinforcement Learning (RL) post-training. The authors demonstrate that VISOR, despite its compact size, achieves strong generalization to unseen environments and provides inherent explainability.

**Strengths:**

- The paper's primary strength lies in its reformulation of navigation from a low-level control problem to a high-level, visually-grounded waypoint selection task. Projecting candidate waypoints as labels onto the panoramic image and having the model reason about which label to choose is a highly innovative approach that elegantly discretizes the action space and naturally integrates with the reasoning capabilities of VLMs.
- The paper makes a compelling case for the "CURE" properties. By creating a unified, single-model agent of a practical size (3B parameters) that eschews external modules like object detectors or segmenters, VISOR presents a more architecturally clean and deployment-friendly alternative to complex, multi-model pipelines.
- The creation of WAYS-Bench is a significant contribution in its own right. As the first dataset designed for supervised fine-tuning of waypoint selection with reasoning traces, it provides a valuable resource for the community and enables a new direction of research in training reasoning-capable navigation agents.

**Weaknesses:**

- The reasoning capability of VISOR during the SFT stage is fundamentally bootstrapped by distillation from a much larger, proprietary model (GPT-4o). This dependency is a critical aspect of the method. The performance is thus capped by the quality of the teacher's reasoning traces, and it's unclear how much of the reasoning ability is "learned" versus "memorized" from these distilled examples.
- The waypoint generation relies on projecting depth data and applying DBSCAN clustering. This crucial preprocessing step is presented as a fixed procedure, but its robustness is not analyzed. It is sensitive to the quality of depth maps, camera parameters, and the choice of clustering hyperparameters, and potential failures in this stage would directly propagate to the policy.
- The results in Table 3 show that VISOR is outperformed by other methods (e.g., MTU3D, Uni-NaVid) on the "Val Seen" splits of the OVON benchmark. While the paper emphasizes generalization, this performance gap suggests that the explicit reasoning and waypoint selection process may be less efficient or direct than other methods in familiar environments.
- All experiments are conducted in the Habitat simulator. While this is a standard and challenging benchmark, the sim-to-real gap remains a major hurdle in robotics. The reliance on panoramic RGB-D data and a shortest-path planner makes direct real-world deployment non-trivial. The absence of any real-world validation is a notable limitation for an embodied AI paper.

**Questions:**

See Weakness

---

> ### Author Response · Authors · 2025-11-30
>
> > **[W1]**: The results in Table 3 show that VISOR is outperformed by other methods (e.g., MTU3D, Uni-NaVid) on the "Val Seen" splits of the OVON benchmark. While the paper emphasizes generalization, this performance gap suggests that the explicit reasoning and waypoint selection process may be less efficient or direct than other methods in familiar environments.
>
> Please refer to the common responses.
>
> ---
>
> > **[W2]**: All experiments are conducted in the Habitat simulator. While this is a standard and challenging benchmark, the sim-to-real gap remains a major hurdle in robotics. The reliance on panoramic RGB-D data and a shortest-path planner makes direct real-world deployment non-trivial. The absence of any real-world validation is a notable limitation for an embodied AI paper.
>
> While real-world robot experiments would enhance impact, they require substantial amount of work, specialized hardware (embodied agents), and non-trivial experimental overhead.
>
> *More importantly, the focus of this work is on the reasoning and decision-making abilities of embodied agents, not their real-world deployment, which is beyond the scope of this venue.*
>
> This choice aligns with standard practice in embodied AI research: the majority of embodied navigation papers report results using the Habitat simulator, for example, [OVON](https://ieeexplore.ieee.org/abstract/document/10802709), [MTU3D](https://openaccess.thecvf.com/content/ICCV2025/papers/Zhu_Move_to_Understand_a_3D_Scene_Bridging_Visual_Grounding_and_ICCV_2025_paper.pdf), and [GOAT-Bench](https://openaccess.thecvf.com/content/CVPR2024/html/Khanna_GOAT-Bench_A_Benchmark_for_Multi-Modal_Lifelong_Navigation_CVPR_2024_paper.html). The Habitat simulator is the de facto standard for language driven object navigation.

---

### Official Review · Reviewer_roEC · 2025-10-31

**Soundness:** 3
**Presentation:** 3
**Contribution:** 2
**Rating:** 4
**Confidence:** 4

**Summary:**

The paper proposes **VISOR**, a ~3B-parameter Vision–Language–Action agent for language-guided object navigation. VISOR reframes high-level control as **selecting a labeled waypoint directly on the panoramic image**, then executing a shortest-path planner to that waypoint. The agent consumes panoramic RGB together with an online top-down map; candidate waypoints are obtained by inverse-projecting depth to valid pixels, clustering them, and overlaying randomly sampled alphanumeric labels on the panorama to enforce image-grounded reasoning. The model produces structured outputs with three tags—`<think>`, `<think summary>`, and `<action>`—to improve interpretability. Training uses supervised fine-tuning on **WAYS-Bench**, followed by RL post-training with **GSPO**, combining a format reward and an action reward that covers both label choice and **Stop**. To mitigate reward hacking, RL relies on a Stop/non-Stop balanced dataset and retains KL regularization. The model is evaluated on **CoIN-Bench** (InstanceObjectNav) and **OVON** (ObjectNav).

**Strengths:**

1. The approach makes it easy for practitioners to understand the model’s outputs and the reasons behind them via structured reasoning tags.
2. The paper introduces waypoint-level supervision to support training reasoning-capable embodied navigation agents, addressing a gap not covered by existing datasets.
3. The method and experimental conclusions are described clearly, making the overall contribution easy to follow.

**Weaknesses:**

1. The paper should clearly argue—ideally with ablations—the necessity of using three cameras/panoramic inputs for the proposed method.
2. The absolute performance lags SOTA. On OVON, for example, *Val Seen* SR: DAgRL 41.3 vs. VISOR-GSPO 21.7; *Val Unseen* SR: MTU3D 40.8 vs. VISOR-GSPO 22.0. On CoIN-Bench, VISOR-GSPO improves over SFT but SR remains modest overall.
3. The paper should quantify the contributions of the `<think>` and `<think summary>` components to final performance, and provide deeper analysis of the respective roles of SFT and RL during training. Based on current results, the **incremental gains from RL** appear limited and warrant further investigation.

**Questions:**

See Weakness

---

> ### Author Response · Authors · 2025-11-30
>
> > **[W1]**: The paper should clearly argue—ideally with ablations—the necessity of using three cameras/panoramic inputs for the proposed method.
>
> As highlighted in our teaser figure (right), we *deliberately* design our observation space to encourage spatial reasoning across multiple images rather than relying on a single RGB frame. A single image provides limited spatial context, making comprehensive reasoning infeasible; in contrast, our multi-image (panoramic) observation space is more aligned with human field-of-view and enables the agent to reason spatially across a broader context.
>
> Using the teaser as an example, the agent is asking “should I go to the kitchen, bathroom or living room?”
> This reasoning process is not possible using a single RGB camera, since the FOV is too limited.
>
> ---
>
> > **[W2]**: The absolute performance lags SOTA. On OVON, for example, Val Seen SR: DAgRL 41.3 vs. VISOR-GSPO 21.7; Val Unseen SR: MTU3D 40.8 vs. VISOR-GSPO 22.0. On CoIN-Bench, VISOR-GSPO improves over SFT but SR remains modest overall.
>
> Please refer to the common responses.
>
> ---
> > **[W3]**: The paper should quantify the contributions of the ```<think>``` and ```<think summary>``` components to final performance, and provide deeper analysis of the respective roles of SFT and RL during training. Based on current results, the incremental gains from RL appear limited and warrant further investigation.
>
> While RL post-training does not substantially improve navigation success rate (SR), it yields a critical benefit, as detailed in row 369: RL training significantly improves navigation efficiency by encouraging the agent to predict the Stop action more frequently and with higher confidence. This refinement directly translates to improved path efficiency, as reflected in improved SPL (Success weighted by Path Length). By optimizing the stopping policy, RL enables the agent to reach goals more decisively without unnecessary exploration, demonstrating that efficiency gains (measured through SPL) are a meaningful contribution to navigation performance beyond raw success metrics.

---

### Official Review · Reviewer_m1Td · 2025-11-01

**Soundness:** 2
**Presentation:** 2
**Contribution:** 2
**Rating:** 4
**Confidence:** 4

**Summary:**

Existing language-driven object navigation methods fall into two categories: one trains VLMs in an end-to-end manner but shows limited generalization and explainability; the other adopts modular zero-shot pipelines but falls short in learning and computation cost. This paper proposes VISOR, a compact, unified, reasoning-capable, and explainable VLA model for the language-driven object navigation task. This 3B model jointly reasons about object recognition and navigation, whose outputs include thinking, thinking summary, and final action. The authors propose WAYS-Bench for the embodied waypoint selection task to SFT Qwen2.5-VL-3B and then perform RL post-training to improve the navigation performance.

**Strengths:**

- Building an intelligent navigation model is a long-standing goal for the community of VLMs and embodied AI. This paper proposes CURE properties that VISOR possesses: compact, unified, reasoning-capable, and explainable.
- Endowing navigation models with thinking capability is a good practice. This facilitates the learning of navigation task and enhances the explainability, and the authors provide some representative qualitative studies and discussions owing to this advantage.
- This paper collects a dataset for the embodied waypoint selection task, which can serve as a data source for the cold start of navigation post-training.

**Weaknesses:**

- I have some concerns about the input modality that make the comparison unfair. VISOR incorporates BEV image as input, which provides global information and could mitigate the challenge of navigation task.
- The results are not strong enough compared to recent navigation models such as Uni-NaVid and MTU3D. For example, using DINO features should not be an excuse, as VISOR leverages pretrained Qwen2.5-VL model in turn.
- Oracle stop can be used for analysis but should not be used for fair comparison. We know that the prediction of stop action is a significant problem in the navigation task. In particular, the oracle stop model cannot even surpass other baselines significantly.
- The improvement from RL post-training is limited. Current bottleneck in navigation appears not to be SFT’s generalization gap, which makes the necessity and advantage of RL not solid.

**Questions:**

- Reward design. I am curious about “training proved difficult, and the model was not able to converge” in Line 1017. So this means, the final results come from the discrete reward while the continuous reward cannot converge? If that is, why?

---

> ### Author Response · Authors · 2025-11-30
>
> > **[W1]**: I have some concerns about the input modality that make the comparison unfair. VISOR incorporates BEV image as input, which provides global information and could mitigate the challenge of navigation task.
>
> We appreciate the reviewer's feedback. We want to clarify a key distinction: our agent does not rely on pre-built maps as input.
> Instead, it constructs maps online using depth information as it navigates.
> The agent begins with an empty map that is progressively populated by the agent in real-time. Note that constructing map online is common practice in the embodied AI field, for example, VLMF (https://ieeexplore.ieee.org/abstract/document/10610712), OpenFMNav (https://aclanthology.org/2024.findings-naacl.24/), TriHelper (https://ieeexplore.ieee.org/abstract/document/10802670), and MTU3D.
>
> Notably, MTU3D generates 2D and 3D features, which are transformed into world coordinates using pose information, creating global queries. Together with an occupancy map, it identifies unexplored regions, deciding if selecting global queries or frontiers.
>
> *Instead, we offer an alternative approach: using the online-built, top-down map as an image, eliciting spatial reasoning in image space. We thus explicitly avoid any form of privileged map information, and instead enforce an active exploration process.*
>
> ---
>
> > **[W2]**: The results are not strong enough compared to recent navigation models such as Uni-NaVid and MTU3D. For example, using DINO features should not be an excuse, as VISOR leverages pretrained Qwen2.5-VL model in turn.
>
> See common responses.
>
> ---
>
> > **[W3]**: Oracle stop can be used for analysis but should not be used for fair comparison. We know that the prediction of stop action is a significant problem in the navigation task. In particular, the oracle stop model cannot even surpass other baselines significantly.
>
> Following standard practice in embodied AI that uses Oracle for performance bounds [for example, IROS 22 (https://ieeexplore.ieee.org/abstract/document/9982216), ICLR 23 (https://openreview.net/forum?id=lTt4KjHSsyl), NIPS22 (https://dl.acm.org/doi/10.5555/3600270.3601450)], we employ an Oracle Stop to establish performance bounds of our VISOR agent.
> Thus,  we compare the performance of our model with and without oracle stop in Tables 2 and 3 of the main paper, and provide more information in Appendix E (line 1025).  By assuming perfect stopping decisions while evaluating the navigation module, we isolate and measure the true contribution of our model, revealing both current performance and the ceiling for improvement through enhanced stopping policies.
>
> ---
>
> > **[W4]**: The improvement from RL post-training is limited. Current bottleneck in navigation appears not to be SFT’s generalization gap, which makes the necessity and advantage of RL not solid.
>
> While RL post-training does not substantially improve navigation success rate (SR), it yields a critical secondary benefit, as detailed in line 369: RL training significantly improves navigation efficiency by encouraging the agent to predict the Stop action more frequently and with higher confidence. This finding help us improving agent’s efficiency, as measured by the improved SPL.
>
> ---
>
> > **[Q1]**: Reward design. I am curious about “training proved difficult, and the model was not able to converge” in Line 1017. So this means, the final results come from the discrete reward while the continuous reward cannot converge? If that is, why?
>
> The design of the reward signal significantly impacts RL convergence. We observed that continuous reward formulations did not converge in our setting, while discrete signals succeeded.
> We speculate this reflects a fundamental trade-off: a strong but coarse gradient signal may facilitate learning better than a weak but fine-grained one. We found support in the DeepSeek-R1 paper [ https://arxiv.org/abs/2501.12948 ]. When the authors introduce a model-based PRM, it inevitably leads to reward hacking (https://proceedings.mlr.press/v202/gao23h.html).
> By using a sparse reward (0/1) on the final answer (with the randomization of the label on the panoramic image), the model has to structure its thinking process to maximize the rewards in getting the right answer. On the other hand, by using continuous reward based on the distance of the label, the reward may become “polluted”, i.e., with noise and with low-variance, and the model may learn to optimize the noise.

---

### Official Review · Reviewer_hxPK · 2025-11-01

**Soundness:** 2
**Presentation:** 3
**Contribution:** 2
**Rating:** 4
**Confidence:** 5

**Summary:**

This paper introduces VISOR, a compact (~3B parameter) Vision-Language-Action (VLA) agent for language-driven object navigation tasks. Unlike prior work that relies on stitched multi-model pipelines or end-to-end embeddings, VISOR performs explicit image-grounded spatial reasoning for both object recognition and action selection, making its policies inherently explainable and generalizable. The method is trained via supervised fine-tuning on the new WAYS-Bench dataset (which provides waypoint-level supervision and reasoning traces) and further refined with a customized RL post-training. The paper provides extensive empirical validation against strong baselines on benchmark tasks, analyzes its strengths and limitations, and details dataset and training procedures.

**Strengths:**

- **Unified and Compact Model:** VISOR is implemented as a single, end-to-end model with 3B parameters, removing the need for large, external object detectors or segmented multi-model pipelines. This directly addresses one of the field's notable practical bottlenecks.
- **Explicit Reasoning and Explainability:** The agent generates detailed reasoning traces (“<think>”, “<think_summary>”, and “<action>”), providing action-level explainability—an advance over black-box action selection models.
- **Dataset Contribution:** The introduction of WAYS-Bench, with waypoint-level supervision and LLM-generated reasoning traces, is a substantive practical and methodological resource for the community. Statistics in Table 1 illustrate dataset diversity.
- **Robust Generalization:** VISOR prioritizes reasoning capabilities that generalize better on “Val Unseen” splits, as shown in Table 2 (CoIN-Bench) and Table 3 (OVON), where performance drops are less severe than conventional policies.
- **Thorough Experimental and Failure Analysis:** The model’s performance is evaluated on multiple challenging benchmarks using standard metrics (SR/SPL), and weaknesses are transparently analyzed with supporting qualitative evidence. The inclusion of ablation and oracle-stop analyses further clarifies algorithmic bottlenecks.
- **Interpretability through Visualization:** Figures (e.g., Figure 1 illustrates spatial reasoning via cluster-based waypoint selection; Figure 2 shows multi-stage, step-wise reasoning in navigation), giving concrete evidence of VISOR’s unique capabilities.

**Weaknesses:**

1. **Empirical Performance Lagging on Seen Categories:**
   - VISOR persistently lags behind the strongest baseline methods (e.g., RL, DAgRL, Uni-NaVid, and MTU3D) in raw performance measures (SPL, SR) on Val Seen and Synonym splits (see Table 3). For example, on OVON Val Seen, VISOR (GSPO) achieves SPL=12.48 / SR=21.7 vs DAgRL’s SPL=21.2 / SR=41.3 and MTU3D’s SPL=23.6 / SR=55.0, indicating that its effectiveness in familiar settings is limited.
   - The rationale for the lower numbers, richer, more flexible natural instructions, and the absence of external modules is acknowledged, but the (frankly, large) gap compromises the perceived impact.
2. **Limited Oral Clarity on Architectural Novelty vs. Prior Reasoning Pipelines:**
   - While the paper positions VISOR against pipeline approaches, the “reasoning traces” paradigm is heavily influenced by prior works (React, PIVOT, ReAct, Goetting et al., 2024; Nasiriany et al., 2024). The main architectural distinction, doing all within a unified smaller model, is important, but the discussion could be stronger on *why/how* the explicit spatial reasoning with panoramic and top-down maps is uniquely beneficial beyond concatenating input modalities and tags. Figure 1, while illustrative, does not make this “wholly new reasoning” distinction sufficiently sharp.
3. **Incomplete Engagement with Key Recent Related Work:**
   - Several directly relevant recent papers are *not* cited or discussed, missing significant context. In particular:
        - *Wen et al., 2024 (VLTNet)*—Directly addresses zero-shot object navigation with VLM-based reasoning, very close to VISOR’s target.
        - *Pan et al., 2023 (LangNav)*; *Yu et al., 2024 (VLN-Game)*—Present alternative vision-language navigation strategies and equilibrium search.
        - *Raychaudhuri et al., 2025 (MLFM)*; *Kuang et al., 2024 (OpenFMNav)*; *Liu, 2024*—Each advances vision-language model-based spatial/semantic map reasoning or open-set navigation.
        - *Zhu et al., 2022*—Provides diagnosis of VLN agent failure modes, which could have strengthened comparative discussion.
     - This incomplete treatment both weakens claims of originality and leaves the reader uncertain if the “CURE” properties are realized elsewhere.
4. **Spatial and Depth Reasoning Remain Weak:**
    - The failure diagnostics in Appendix C and supporting visualizations (Figures 5, 6, 8) show systematic mistakes: hallucination of labels (incorrect spatial placement or scene element), left-right confusion, and poor depth estimation. The “Markovian Setup” problem is described in the text and visible in Figure 7, but little is done to ablate or mitigate these weaknesses. Without memory/history, the agent is prone to errors in last-mile navigation, where feature-based policies often excel.
5. **Insufficient Ablation on Component Contributions:**
   - The impact of reasoning traces, panoramic input, and top-down map is not individually ablated. Readers are left wondering which design choices are responsible for generalization gains, and whether, for example, simpler memory or attention augmentations could close the gap with less explainable policies.
6. **Mathematical Detail Only at the Level of Standard RL Loss and SFT; No Tight Model-Reasoning Formalization:**
   - The loss functions (Equation for $\mathcal{L}_{\mathrm{SFT}}$ and GSPO) are standard in supervised/sequence RL, and not unique to VISOR. The paper does not formalize the reasoning traces process as an algorithmic/optimization object beyond the input-output prompt structure. There is no clear mathematical framework connecting the agent's reasoning outputs to improved sample efficiency or generalization. This is particularly relevant since a key claim of the paper is about “human-like reasoning.” A reader is likely to expect tighter equations or formal reasoning for *why* this reasoning enables improved performance or explainability.
7. **Dataset Collection (“Ground-truth Reasoning”) Blurring Human and LLM Roles:**
   - The use of GPT-4o to retrospectively generate ground-truth reasoning traces for SFT may not always result in authentic or diverse human-like strategies (see Figure 3 for sample traces). This proxy could introduce bias: if the model simply learns to mimic LLM-generated justifications that do not always match the semantic nuances required for true generalization, this could cap performance.
8. **Limited Analysis of Efficiency/E2E Latency:**
   - While VISOR is more efficient and compact than stitched pipelines, there is no explicit timing analysis for model inference speed, nor is there a clear accounting of compute/latency improvements over strong modular or larger baselines. This undermines some claims around “practical efficiency.”
9. **Minor:**
   - Some figures and tables, e.g., Figure 2 and Figure 3, would benefit from clearer legends and color coding, as it can be challenging to align the visualization with textual step descriptions for those unfamiliar with the setup.
   - Occasional typos and grammatical slips (e.g., “efficixmally” in system prompt snippet) in the Appendix/prompt code.

## Potentially Missing Related Work

1. Wen, C., Huang, Y., Huang, H. (2024): Zero-shot Object Navigation with Vision-Language Models Reasoning. Close conceptual overlap: VLTNet integrates vision-language and reasoning for zero-shot navigation; should be cited and compared in Related Work and compared as an alternative in Results.
2. Pan, B., Panda, R., Jin, S. (2023): LangNav: Language as a Perceptual Representation for Navigation. Directly relevant to language-driven navigation; should be discussed in Section 5 and cited after initial task framing.
3. Yu, B., Liu, Y., Han, L. (2024): VLN-Game: Vision-Language Equilibrium Search for Zero-Shot Semantic Navigation. Alternative zero-shot navigation framework; should be noted as baseline or related comparison.
4. Raychaudhuri, S., Cancelli, E., Campari, T. (2025): MLFM: Multi-Layered Feature Maps for Richer Language Understanding in Zero-Shot Semantic Navigation. Presents attribute/relation-centric navigation; discuss as a complementary approach in Related Work.
5. Kuang, Y., Lin, H., Jiang, M. (2024): OpenFMNav: Towards Open-Set Zero-Shot Object Navigation via Vision-Language Foundation Models. Shares goals of generalization; compare and discuss weaknesses/strengths.
6. Zhu, W., Qi, Y., Narayana, P. (2022): Diagnosing Vision-and-Language Navigation: What Really Matters. Provides insight into failure modes, valuable for failure analysis and benchmarking.
7. Liu, W. (2024): Towards Vision and Language Models Aided Object Navigation. Hierarchical vision-language navigation; should be discussed in Section 5 as a relevant recent effort.
8. Stone, A., Xiao, T., Lu, Y. (2023): Open-World Object Manipulation using Pre-Trained Vision-Language Models. For positioning in relation to use of pre-trained VLMs.

Each should be integrated into the literature review (Section 5) and, where appropriate, Experimental Results and Failure Analysis, to sharpen the paper’s conceptual originality and positioning.

**Questions:**

- Can the authors clarify the unique gains attributable to each architectural element: panoramic view, top-down map, and explicit reasoning tags, via ablation studies, or, at the very least, a detailed qualitative analysis?
- How does the use of LLM-generated (rather than human-generated) reasoning traces as “ground-truth” affect generalization? Any empirical evidence that agents trained on such traces adopt meaningful human spatial strategies, especially in ambiguous instructions?
- Given systematic spatial and depth perception errors (Figure 6, Figure 8, repeated left-right confusion), what candidate approaches might address these issues? Are there plans to fuse in memory/history, or would hybrid (learned + symbolic) representations help?
- How much does the explicit “reasoning” in <think> and <think_summary> actually contribute to policy improvement versus simply providing post-hoc explainability?
- Can the authors report wall-time and GPU-budget comparison against leading multi-model pipelines, especially those that use larger LLMs or external object detectors (e.g., MTU3D, Uni-NaVid)? Is VISOR’s claimed efficiency realized in practice, or are savings offset by prompt engineering and additional reasoning outputs?
- What plans exist for systematically evaluating the credibility and diversity of reasoning traces provided by the agent versus human raters?
- Are there meaningful limitations in transferring VISOR to real-world robotics or other simulators, especially regarding depth, occlusion, or physically dynamic environments?

---

> ### Author Response · Authors · 2025-11-30
>
> > **[W1]**: Empirical Performance Lagging on Seen Categories
>
> Please refer to the common responses.
>
> ---
>
> > **[W2]**: Limited Oral Clarity on Architectural Novelty vs. Prior Reasoning Pipelines.
>
> We thank the reviewer, we will revise the main paper to be more clear about the distinction.
> As highlighted in our teaser figure (right), we deliberately design our observation space to encourage spatial reasoning across multiple images (panorama + top-down) rather than relying on a single, isolated RGB frame. A single image provides limited spatial context, making comprehensive reasoning infeasible; in contrast, our multi-image (panoramic) observation space is more aligned with human field-of-view and enables the agent to reason spatially across a broader context (in the teaser example, “should I go to the kitchen, bathroom or living room?”).
>
> Moreover, differently from the other, we distill this reasoning in a small model (3B), following the cure properties.
>
> ---
>
> > **[W3]**: Incomplete Engagement with Key Recent Related Work.
>
> We kindly *disagree* about the incomplete related work section, and that “leaves the reader uncertain if the “CURE” properties are realized elsewhere.” In particular:
>
> - Towards Vision and Language Models Aided Object Navigation **appears to be a Master thesis**, and we don't find the work relevant to be discussed.
>
> - VLTNet addresses the Language Driven Zero-shot Object Navigation (L-ZSON). As written in the main paper (row 304-305), we compare against InstanceObjectNav and ObjectNav **training-based** policies, following the CURE properties. Moreover, VLTNet is composed of a modular pipeline, namely an object detector (GLIP), an external LLM for frontier selection coupled with frontier based exploration and GPT-3.5 for goal identification. Comparison is not possible due to the code not being available.
>
> - LangNav proposes to use language as a perceptual representation for the vision-and-language (VLN) task. InstanceObjNav and ObjNav are considered orthogonal to VLN (unless proposing a generalist agent); therefore, we do not find LangNav essential for this work.
>
> - VLN-Game proposed a zero-shot agent for semantic navigation: during navigation, it uses an object detector and class-agnostic segmentation module to create a 3D map (again, note that this agent does not follow our proposed CURE properties).
>
> - Similarly, OpenFMNav proposes a multi-module pipeline: it uses GroundedSAM for detection and segmentation, and GPT4-V for discovering objects in images.
>
> - Diagnosing Vision-and-Language Navigation proposes valuable insight into failure mode of trained agents on the VLN task. However, the considered task (VLN) requires agents to follow detailed and long instruction, which is different from our Language-driven object navigation. Thus, the insights from the paper are not directly generalizable to this setup.
>
> ---
>
> > **[W4]**: Spatial and Depth Reasoning Remain Weak.
>
> We have thoughtfully identified key limitations (Section 4.3) and outlined future work (Conclusion) to address them systematically. We believe each of these limitations could open several interesting research directions for embodied reasoning:
>
> - History-augmented reasoning could enable non-Markovian policies with stronger learning signals (e.g., multi-image reasoning, trajectory reasoning).
>
> - Depth-enriched spatial maps could provide finer-grained geometric understanding. For example, projecting depth information, such as a cone representing the “meters” of depth, into the top-down map could present an interesting challenge for improving depth estimation using multi-image reasoning (panoramas + top-down depth representations).
>
> - Hallucination-aware rewards could leverage RL to penalize unreliable predictions.
>
> ---
>
> > **[W5]**: No Tight Model-Reasoning Formalization.
>
> We appreciate this feedback, but respectfully disagree that formal mathematical proofs are necessary or appropriate for this work.
>
> First, literature demonstrates that reasoning and RL improve performance and generalization. However, understanding "why" remains an open question in the field, currently only speculated upon and not formally proven. *A formal derivation of why reasoning improves performance is beyond the scope of this paper and arguably represents a separate research contribution in interpretability/RL theory.*
>
> Second, explainability does not require mathematical formalism. Our method's strength lies in providing transparent, interpretable reasoning traces, i.e. the agent explicitly explains the rationale behind each navigation decision through natural language. This constitutes genuine explainability without requiring mathematical formalization.
> Our contribution is demonstrating that reasoning-based agents can be built end-to-end and produce interpretable outputs. The theoretical foundations of why this works remain an important open question, but one orthogonal to validating the approach itself.

---

> > ### Author Response · Authors · 2025-11-30
> >
> > > **[W6]**: Dataset Collection (“Ground-truth Reasoning”) Blurring Human and LLM Roles.
> >
> > While we appreciate the suggestion, collecting a dataset with human annotations is prohibitively costly and practically unfeasible for this work. Large-scale annotation requires substantial resources, recruiting and training annotators, quality control, and significant time investment. This overhead is incompatible with the scope and timeline of this project.
> >
> > ---
> >
> > > **[W7]**: Are there meaningful limitations in transferring VISOR to real-world robotics or other simulators, especially regarding depth, occlusion, or physically dynamic environments?
> >
> > Regarding physically dynamic environments, we acknowledge that our map representation is static and not updated during navigation. However, this limitation is shared by most current map-based navigation methods and is not a fundamental constraint of our approach.
> >
> > The simulator imposes no restrictions on sensor configuration or computational overhead, meaning that incorporating additional RGB or depth sensors is straightforward.
> >
> > For real-world deployment, the primary requirement is the camera configuration. As stated in Footnote 1 (row 161), *“This setup can be readily simulated on humanoid robots or other embodied agents via head rotation.”* Thus, our panoramic observation model is compatible with others embodied platforms.

---

### Author Response · Authors · 2025-11-30
**Common response**

> Reviewer roEC - The absolute performance lags SOTA. On OVON, for example, Val Seen SR: DAgRL 41.3 vs. VISOR-GSPO 21.7; Val Unseen SR: MTU3D 40.8 vs. VISOR-GSPO 22.0. On CoIN-Bench, VISOR-GSPO improves over SFT but SR remains modest overall.

> Reviewer m1Td - The results are not strong enough compared to recent navigation models such as Uni-NaVid and MTU3D. For example, using DINO features should not be an excuse, as VISOR leverages pretrained Qwen2.5-VL model in turn.

> Reviewer oKXw - The results in Table 3 show that VISOR is outperformed by other methods (e.g., MTU3D, Uni-NaVid) on the "Val Seen" splits of the OVON benchmark. While the paper emphasizes generalization, this performance gap suggests that the explicit reasoning and waypoint selection process may be less efficient or direct than other methods in familiar environments.

> Reviewer hxPK - Empirical Performance Lagging on Seen Categories

While zero-shot composite pipelines leverage state-of-the-art modules and often achieve competitive results, they suffer from fundamental limitations, as discussed in our introduction and abstract. In contrast, this work prioritizes end-to-end learning for navigation.


Notably, Uni-Navid it’s a 7B LLM, while MTU3D also leverages SOTA components (i.e., Fast-SAM and DINO) to construct segment-level representations that improve navigation performance (MTU3D paper, Section 3.1).

Similarly, [OVON](https://ieeexplore.ieee.org/abstract/document/10802709) showed that equipping its DagRL policy with an object detector *significantly* improved results. Specifically, Table 3 in the OVON paper shows that on the Val Unseen split, SR increases from 18.3→37.1, and in Val Seen Synonyms SR increases from 14.4 → 21.4, *further validating that the usage of SOTA models substantially helps navigation performance.*

**However, in this paper, as also noted by the reviewers, we propose a compelling alternative, i.e., an agent that follows our proposed CURE properties: being Compact (3B), Unified, Reasoning-capable and explainable. Embodied agents should be able to do navigation E2E, without relying on modular pipelines.**

Moreover, this paper prioritizes explainability, spatial reasoning, and end-to-end learning over absolute peak performance.
**Critically, the relevant metric for evaluating navigation agents is generalization to unseen environments, not performance on seen categories data.**

While we acknowledge lower performance on Val Seen compared to DAgRL (41.3 vs. VISOR-GSPO 21.7), the Val Unseen split is the only meaningful evaluation.
Here, the performance gap is in our favor: DAgRL 18.30 vs. VISOR-GSPO 22.0.
More importantly, this reveals a fundamental difference in generalization:
- DAgRL: 41.30 (Val Seen) → 18.30 (Val Unseen)  → large performance drop
- MTU3D: 55 (Val Seen) → 40.80 (Val Unseen) → significant drop
- VISOR-GSPO: 21.7 (Val Seen) → 22.0 (Val Unseen) → consistent performance across split.

Our method demonstrates robust generalization without the large performance cliff observed in competing methods. This consistency, i.e., maintaining performance across seen and unseen environments is a stronger indicator of a learned navigation policy than performance on familiar scenes.

---

### Author Response · Authors · 2025-11-30
**Summary of Strengths (as acknowledged by reviewers)**

We thank the reviewers for their comments. Below is a summary of their feedback on our work.

Reviewer oKXw highlights our task reformulation, noting that “[…] reasoning about which label to choose is **a highly innovative approach**.” The paper presents a compelling case for the CURE (compact, unified, reasoning-capable, and explainable) properties, which offer a more architecturally clean solution compared to multi-model pipelines. Reviewer m1Td also emphasizes the importance of CURE properties, describing them as a long-standing goal in embodied AI.

Our methods and experimental conclusions are clearly described (roEC), and the **figures provide concrete evidence of our agent VISOR’s unique capabilities** (hxPK). Additionally, our proposed dataset **WAYS-Bench is a significant contribution**, offering a valuable resource for the community and opening new directions for research (oKXw). Reviewer hxPK calls WAYS-Bench **“a substantive practical and methodological resource for the community.”**

Finally, our agent VISOR demonstrates strong generalization (hxPK), provides action-level explainability, and represents an advance over black-box models, making it easier for practitioners to understand the model’s outputs (hxPK, roEC).

---

### Meta-Review · Area_Chair_tSES · 2026-01-04

**Summary:**

The paper introduces VISOR, a 3B-parameter Vision-Language-Action (VLA) agent designed for language-driven object navigation. The core contribution is a unified architecture that replaces traditional multi-model pipelines with explicit, image-grounded reasoning traces ("think", "think summary", and "action"). Although the authors have addressed some of the concerns raised by reviewers, the major concern regarding its huge performance drop still persist. For example, as raised by reviewer hxPK

> VISOR persistently lags behind the strongest baseline methods (e.g., RL, DAgRL, Uni-NaVid, and MTU3D) in raw performance measures (SPL, SR) on Val Seen and Synonym splits (see Table 3). For example, on OVON Val Seen, VISOR (GSPO) achieves SPL=12.48 / SR=21.7 vs DAgRL’s SPL=21.2 / SR=41.3 and MTU3D’s SPL=23.6 / SR=55.0, indicating that its effectiveness in familiar settings is limited.

For a method to be applicable in real-world scenarios, we expect its performance is consistent with respect to both seen and unseen scenarios. I feel this is paper has a potential, but still got some room for improvement.

**Reviewer Concerns:**

Concerns that have been addressed
- concerns about the input modality
- potential related works

Concerns not addressed
- performance drop on val seen
- quantitative results about the contributions of the <think> and <think summary> components

**Reviewer Scores:**

I think all reviewers might not change their scores, because the shared concerns about the performance drop and additional quantitative results are not addressed during rebuttal.

---

### Decision · Program_Chairs · 2026-01-26

Reject